

# Simulated Long-term Evolution of the Thermosphere during the Holocene: 2. Circulation and Solar Tides

Xu Zhou[1,3], Xinan Yue[1,2,3], Yihui Cai[1,2,3], Zhipeng Ren[1,2,3], Yong Wei[1,2,3], Yongxin Pan[1,2,3]

[1]Key Laboratory of Earth and Planetary Physics, Institute of Geology and Geophysics, Chinese Academy of Sciences, Beijing, 100029, China

[2]College of Earth and Planetary Sciences, University of Chinese Academy of Sciences, Beijing, 100029, China

[3]Beijing National Observatory of Space Environment, Institute of Geology and Geophysics, Chinese Academy of Sciences, Beijing, 100029, China

*Correspondence to*: Xinan Yue (yuexinan@mail.iggcas.ac.cn)

**Abstract.** On timescales longer than the solar cycle, long-term changes in CO2 concentration and geomagnetic field have the potential to affect thermospheric dynamics. In this paper, we investigate the thermospheric dynamical response to these two factors during the Holocene, using two sets of ~12,000-yr control runs by the coupled thermosphere-ionosphere model, GCITEM-IGGCAS. The main results indicate that increased/decreased CO2 will enhance/weaken the thermospheric circulation throughout the Holocene, but this effect is nonlinear. The cooling effect of CO2 in the thermosphere further provides plausible conditions for atmospheric tidal propagation and increases the thermospheric tidal amplitude. Geomagnetic variations induce hemispheric asymmetrical responses in the thermospheric circulation. Large changes in the circulation occur at high latitudes in the hemisphere with distant magnetic poles drift, inferring a crucial role of geomagnetic non-dipole variations in circulation changes. A positive correlation between the diurnal migrating tide (DW1) and geomagnetic dipole moment is revealed for the first time. The amplitude of DW1 in temperature will increase by ~1–3 K for each $1\times10^{22}$ Am$^2$ increase in dipole moment.

## 1 Introduction

The main external energy input to the terrestrial thermosphere is solar radiation, particularly in the extreme ultraviolet (EUV) band. The solar-driven circulation manifests as the flow across the isobars, in contrast to the geostrophic flow that dominates in the middle and lower atmosphere (Forbes, 2007). This is because the Coriolis force is much smaller than the pressure gradient term for the typical terrestrial thermosphere. Under absorption of solar daily-cyclic forcing, the atmosphere also induces the solar tides, which refers to global-scale perturbations in atmospheric parameters with periods and zonal wave numbers that are harmonics of a day and a zonal cycle. In addition to the local absorption of EUV radiation as the major source, the solar tides in the thermosphere also come from upward propagating waves excited in the middle and lower atmosphere, including the infrared absorption by tropospheric H2O and ultraviolet absorption by stratospheric O3 (Forbes and Zhang, 2022). Thus, the level of solar activity is expected to have a key impact on the dynamical variability in the thermosphere



(Oberheide et al., 2009; Sun et al., 2022). However, when inspecting on time scales longer than the solar cycle, the influence
from other secular variables, such as long-term changes in $CO_2$ concentration and main geomagnetic fields, should not be
ignored. It is then natural to ask how and to what extent these factors act on the thermospheric dynamics on long-term time
scales, e.g., since the Holocene.

$CO_2$ plays a significant role in cooling the thermosphere, in contrast to the warming effect in the troposphere (Laštovička et
al., 2006; Solomon et al., 2018). Since the first prediction by Roble & Dickinson (1989), many observational evidences and
simulation experiments have been subsequently proposed to support the $CO_2$ cooling effect using modern techniques and
advanced models (Akmaev & Fomichev, 2000; Akmaev et al., 2006; Marsh et al., 2013; Ogawa et al., 2014; Qian et al., 2011;
2006; Solomon et al., 2015; Zhang et al. 2016). A well-established consensus is that every 10 ppm increase in $CO_2$
concentration will result in a ~1–3K decrease in global-mean temperature in the thermosphere (e.g. Solomon et al., 2018). As
the issue of increasing $CO_2$ becomes urgent (IPCC, 2014), have also worked to elucidate the concomitant effects on the upper
atmosphere (Zhou et al., 2022), and one of which is the thermospheric dynamics. Using the GAIA (Ground-to-topside
Atmosphere Ionosphere model for Aeronomy) simulation, Liu et al. (2020) suggested that the doubling of $CO_2$ concentration
should strengthen thermospheric meridional circulation, enhance diurnal migrating tide, and weaken semidiurnal migrating
tide. Kogure et al. (2022) further analyzed the underlying mechanism of the thermospheric zonal mean wind response,
suggesting that the ion drag, molecular viscosity, and meridional pressure gradient forces as the three main factors are in the
combined modulation. However, the impact of $CO_2$ on the long-term evolution of the thermospheric dynamics during the
Holocene is still poorly inderstood.

The secular variation of the geomagnetic field would produce considerable changes in the thermosphere temperature other
than the $CO_2$ effect. Although the geomagnetic variation does not act directly on the neutral atmosphere, it affects ion motion
and thus ionospheric behavior (Cai et al., 2019; Elias et al., 2022; Yue et al., 2018; Zossi et al., 2018), which are coupled to
the neutral atmosphere via ion-neutral collisions. The strength of the geomagnetic field determines the gyrofrequency and the
ionospheric conductivity, thus influencing the Joule heating power and $\mathbf{E}\times\mathbf{B}$ drift velocities (Cnossen et al., 2012; Zhou et al.,
2021). The geomagnetic tiled angle controlling the geographic distribution of the Joule heating should produce further changes
in temperature and neutral winds (Cnossen & Richmond, 2012). Cnossen (2014) reported that the geomagnetic variation over
the last century could cause a ~±10K change in the thermosphere temperature regionally, comparable to the –8K decrease in
global temperature due to increased $CO_2$ over the same period. Analyses of recent decades (Cnossen et al., 2020) and
projections in the coming decades (Cnossen et al., 2022) about the thermospheric climate change confirm the importance of
the geomagnetic variation, although accelerating $CO_2$ growth still plays a dominant role. Since the geomagnetic field has





undergone a more complex evolution during the Holocene than in the present century (Korte et al., 2011), the impact on the
evolution of thermospheric dynamics is expected to be more dramatic and therefore worth investigating.

The aim of the present study is to discuss the scenario of thermospheric dynamic changes due to the long-term changes in CO2
concentration and geomagnetic field during the Holocene. This paper is organized as follows: Section 2 will briefly introduce
the numerical simulation settings. Section 3 will show the main results of the simulations, then Section 4 discuss the scientific
key points. In the end, a short summary is given in Section 5.
**2 Model Description and Settings**
Attempting to understand the long-term evolution of thermospheric dynamics affected by these two factors in the Holocene,
we designed long-term time-slice simulations based on the Global Coupled Ionosphere-Thermosphere-Electrodynamics Model
developed at the Institute of Geology and Geophysics, Chinese Academy of Sciences (GCITEM-IGGCAS, Ren et al., 2009,
2010, 2011, 2020). Detailed model description and settings are referred to Yue et al. (2022) and Cai et al. (2023), which have
carefully investigated the global thermal structure and density profile of the thermosphere and ionosphere, respectively. Here,
we give a briefly introduction to restate and to add key information. This 3-dimensional coupled thermosphere-ionosphere
model self-consistently solves the global thermospheric and ionospheric behavior in the altitudinal coordinate, covering
altitudes from 90 km to 600 km. The ionospheric electro-dynamics is solved on the provided geomagnetic field configuration
using magnetic apex coordinates (Richmond, 1995) based on a set of spherical harmonic coefficients. The calculation scheme
requires the geomagnetic field to be dipole-dominated, so the situation of geomagnetic reversal is difficult to portray. The
high-latitude electric potential and electric fields are specified by the empirical model of Weimer-96 (Weimer, 1996), which
is driven by the hemispheric power (HP), solar wind speed (SWS), interplanetary magnetic field (IMF), and cross-polar cap
potential (CPCP). At the lower boundary at 90 km, migrating tide in neutral temperature and density are given by the Global
Scale Wave Model (GSWM), while neutral winds are self-consistently calculated. Non-migrating tides are not included in this
study. The solar EUV radiation is described by the empirical model EUVAC (Richards et al., 1994), which is driven by the
proxy of solar flux at 10.7 cm (F10.7). The $CO_2$ cooling is calculated under the assumption of the nonlocal thermodynamic
equilibrium (NLTE) with a cooling-to-space approximation assumed. In this model, the $CO_2$ level is specified by a given value
for a fixed time under the assumption of diffusive equilibrium. This calculation formula follows Roble et al. (1988), and is
also adopted by other thermosphere-ionosphere coupled models, such as NCAR-TIEGCM (Qian et al., 2017).

To diagnose the long-term effects of $CO_2$ and geomagnetic field variations on the thermospheric dynamics, two control runs
(CR1 and CR2) were performed under perpetual solar minimum and geomagnetic quiet condition, which correspond to the



CR2 and CR1 in the Yue et al. (2022) and Cai et al. (2023). The driving parameters in Weimer-96 model are set as HP =
10GW, SWS = 300 m/s, IMF By = 0 nT, IMF Bz = –0.5 nT, and CPCP = 20 kV for both cases, representing the extreme
geomagnetically quiet condition of Kp = ~0.3. To eliminate the impact of solar variation, each case was performed under solar
minimum, correspondingly the F10.7 setting to be constant of 87sfu (solar flux unit, 1 sfu = $10^{-22}$ W m$^{-2}$ Hz$^{-1}$). In CR1, realistic
$CO_2$ from a combined dataset drives the GCITEM-IGGCAS model with a fixed configuration of geomagnetic fields. Hence,
the simulated variability of the thermosphere is derived exclusively from the $CO_2$ changes. The $CO_2$ dataset consists of three
components: (1) Estimation from the ice cores recorded air composition in the interval of ~80,000 yrs before ~1650 with a
rough resolution of ~100 yrs during the Holocene (Lüthi et al., 2008). (2) Measurement in ice with high precision back to 2000
yrs before the present (MacFarling Meure et al., 2006). (3) Modern atmospheric measurement at Mauna Loa Observatory,
Hawaii, since 1958 (Keeling et al., 1995). In CR2, the $CO_2$ level is fixed to be 270 ppm, corresponding to the averaged value
during the Holocene, while the geomagnetic fields are set to be varied with time. The specified geomagnetic field before 1900
is provided by the CALS10k.2 model developed by Constable et al. (2016), which is based on the archeo-magnetic and lake
sediment data. Generally, this model roughly has spherical harmonics to degree and order of 10, and cubic B-splines
parameterization is implemented with knots positioned every 40 yrs. After 1900, the geomagnetic fields are described by the
International Geomagnetic Reference Field (IGRF) model (Alken et al., 2021). This model is based on the modern magnetic
observations to describe the spatial distribution of geomagnetic fields by the spherical harmonic degree and order of 13 with
the time resolution of 5 yrs. Both cases were run every 100 yrs in the period of 9455 BC to 1945 AD, and an additional run of
2015 AD was for the contemporary condition. Particularly, pre-runs of 15 days were performed as spin-up preparation to
eliminate the influence from the initial conditions, and the outputs in the last day were used for analysis. Each case was running
in two seasons, March and June, with the aim of discussing the seasonal dependence of the thermospheric dynamical response..

## 3 Results

### 3.1 CO2 effect

According to the CR1 results, Figure 1 illustrates the changes in zonal-mean winds due to increased $CO_2$ from 1945 to 2015
(310 to 400 ppm), exemplifying how the changes in $CO_2$ act on the thermospheric circulation. Figures 1b–1d show the
strengthening of the thermospheric circulation in March, mainly including enhanced equatorward flow from the north and
south poles, accelerated eastward flow at mid- and low-latitudes latitudes, and increased downward/upward movement in the
upper/lower thermosphere. The acceleration of the eastward zonal and equatorial meridional winds winds is about ~1–2 m/s
when $CO_2$ is increased by ~90 ppm. The $CO_2$ acceleration effect of the thermosphreic circulation is also evident in June.
Figures 1f–1h show the enhanced summer-to-winter prevailing wind and corresponding increased westward/eastward zonal
wind in the summer/winter hemisphere due to the Coriolis force. The vertical winds also show a downward increase in the





upper thermosphere, while the slight increase in the lower thermosphere disappears around the winter pole. Compared to the
wind change in March, the accelerated thermospheric winds in June achieve ~2–3 m/s in zonal and meridional, and a few cm/s
in vertical. Our simulation gives a reasonable and convincing result compared to the GAIA simulation of Liu et al. (2020),
which shows an increase in the meridional winds of 5–15 m/s when $CO_2$ increases by 345 ppm.

Examining the $CO_2$ effect on the thermospheric circulation throughout the Holocene, Figure 2 illustrates the time evolution of
changes in meridional wind versus latitudes in the CR2 simulation. The chosen height of ~197 km is where the changes in the
meridional wind are significant as shown in Figure 1. The result for the beginning year (–9455) have been subtracted in order
to show the $CO_2$ effect more intuitively. The corresponding $CO_2$ variation is plotted in red-solid line, which is also subtracted
the $CO_2$ level in the beginning year (264 ppm). Changes in the meridional circulation are obviously highly correlated with
$CO_2$ variation, and become much more significant since ~1800 when the increase in $CO_2$ was much larger due to the industrial
revolution. The correlation coefficient is generally over ±0.99 at most latitudes. During the equinox season, the meridional
circulation varied to be much equatorward/poleward due to the increase/decrease of $CO_2$. As for the solstice season, the $CO_2$
effect manifests to be acceleration/deceleration of the summer-to-pole circulation. For the past over 10,000 years before ~1800,
the change in meridional circulation velocity in March and June only fluctuated by ∓0.4–±0.1 m/s and –0.6–−0.2 m/s,
respectively. However, in the last 200 years, the $CO_2$-induced changes in meridional wind could reach more than 1 m/s. Figure
3 further analyses the $CO_2$ effect on the thermospheric dynamics, choosing the averaged zonal circulation as a proxy. The
results show that $CO_2$ enhances the eastward flow at the equator during March, rather than being strictly linear. The growth
of the accelerated eastward flow becomes small as $CO_2$ increases. Linear regressions show a change of 0.012 m/s in the
thermospheric equatorial zonal flow per ppm $CO_2$ increase, and the parabolic fit should be in good agreement with the
simulated data. The parabolic fitting obviously indicates that the rate of change of the thermospheric circulation slows down
at the present $CO_2$ level. A similar nonlinear effect is also manifested in the June zonal circulation (Figure 3c).

As for the solar tidal response to the $CO_2$ variation during the Holocene, Figure 4 illustrates the time evolution of diurnal
migrating tide in temperature (DW1-T) at ~240 km, which is the major tidal component in the thermosphere. The DW1-T tidal
amplitude is positively correlated with $CO_2$ changes, manifesting as increasing by ~10 K compared with the beginning year
(-9455) during March when the $CO_2$ level achieve 400 ppm in the modern era, particularly maximizing at the equatorial and
low-latitude region. From 8000 BC to 4000 BC, when the $CO_2$ level was low throughout the Holocene, the DW1-T amplitude
also decreased slightly. The specified DW1-T amplitude at the lower boundary in March is a maximum of ~16 K at the equator
and two secondary peaks of ~7 K at ±35°. As for the DW1-T at the lower boundary in June, the strength is about ~1/2 of that
during March. Correspondingly, the changes in the thermospheric DW1-T amplitude in the modern era are slightly over 2 K,



only ~1/4 than that in March. The maximum change is found at mid-latitudes in the winter hemisphere, rather than the equator.
The latitudinal difference in the DW1-T changes is contrary with the DW1-T time tendency, which generally maximizes in
the summer hemisphere (Gu & Du, 2018).
**3.2 Geomagnetic field effect**
The geomagnetic field effect on the thermospheric circulation is regional and complicated, unlike the global effect of $CO_2$.
Figure 5 exemplified the thermospheric circulation in the present era in the CR2 simulation, and manifested how the circulation
changed over the past 70 years due to the geomagnetic variation. The thermospheric winds generally flow across the isotherm
due to the pressure gradient force and can maximize over 100 m/s around the terminator. The auroral heating modulates the
solar-driven winds and decreases the poleward flow at high- and mid-latitudes. Figure 5b shows that the geomagnetic variation
from 1945 to 2015 alters the geographic distribution of temperature in March, notably at high latitudes (~±15 K) and not
negligibly at mid- and low-latitudes (±5 K). Correspondingly, the change in horizontal neutral winds could exceed 30 m/s at
high latitudes and around the dusk sector. The changes in temperature and wind induced by the geomagnetic field are smaller
in June than that in March, which is about ±10 K/±3 K at high/mid-low latitudes for temperature and maximizes ~20 m/s for
horizontal winds. The circulation change in the northern hemisphere is much larger in the southern hemisphere, regardless
during March or June. The horizontal wind changes in the southern hemisphere are generally smaller by 10–20 m/s than that
in the northern hemisphere, and the temperature change is smaller by 5–10 K. The hemisphere difference is coincident with
the asymmetrical change in the geomagnetic poles. The northern magnetic pole shifted 12° and 76° in latitude and longitude,
respectively. However, the southern magnetic pole drifted by merely 4° and 7° in latitude and longitude, respectively.

In addition, Figures 5b and 5d show that the geomagnetic variation during the period 1945–2015 induced different temperature
responses during the daytime/nighttime at mid- and low-latitudes. This local-time-dependent effect is further examined in
Figure 6 and Figure 7 for the month of March and June, respectively. Figure 6a illustrates the local-time dependence of
temperature changes due to the geomagnetic variation with respect to the beginning year of 9455 BC, when the dipole moment
of the geomagnetic field underwent a minimum period. During the daytime, the average temperature at low-latitude was
generally higher than in 9455 BC for most of the time, except for 4900 BC and 4700 BC. The changed magnitude varied from
–2 K to 9 K. In contrast, the nighttime temperature change is negative compared to 9455 BC since 3100 BC, and ranges from
–7 K to +6K before 3100 BC. We then deduced the day-night differences in the temperature response at mid- and low-latitudes
and illustrated them in comparison with the strength of the geomagnetic dipole moment in Figure 6b. The results show an
obviously positive correlation between the day-night differences and the geomagnetic dipole moment, indicating that a stronger
geomagnetic dipole moment would induce larger day-night temperature differences in the thermosphere at mid-to-low latitudes





in March, thereby exacerbating the prevailing day-to-night flow. During the whole simulation period in the Holocene, the day-
night difference in temperature caused by the geomagnetic variation can vary up to ~15–20K. The fluctuation magnitude is
about 5% concerning the day-night temperature difference in the thermosphere is generally 300–400K. Meanwhile, the
geomagnetic dipole moment varies more than 40%. As for the case of June, the positive correlation is not valid for all latitudes
and becomes more complicated. As the dipole moment increases, the average temperature at low-latitudes decreases for both
daytime and nighttime. The change in the day-night temperature difference is weaker than that in March. Around the equator
and in the southern mid-latitudes, the day-night difference in temperature decreases while the geomagnetic dipole moment
increases, such as during 8000–6600 BC and 2600 BC–1600 AD.

As mentioned above, the daytime temperature responses in the thermosphere differed from that of the nighttime due to the
geomagnetic variation, suggesting that the tidal response should also be affected, especially during March. Figure 8 then
examines the thermospheric tidal response to the geomagnetic variation during the Holocene in the CR2 simulation, including
the diurnal and semidiurnal migrating tides in temperature (DW1-T and SW2-T). These two major tidal components respond
differently to the geomagnetic variation. The strength of DW1-T is positively correlated with the geomagnetic dipole moment.
When the dipole moment intensity becomes ~40% larger than at the beginning of the simulation, the amplitude of DW1-T
increases correspondingly by ~10 K. However, the SW2-T around the equator is negatively correlated to the geomagnetic
dipole moment, while at mid-latitudes it is positively correlated. The strength of SW2-T response to the geomagnetic variation
is much smaller than that of DW1-T, and ranges within ~±2K throughout the simulation period in the Holocene. Figure 9
further diagnoses the relationship between the thermospheric migrating tides and the geomagnetic dipole moment for different
thermospheric altitudes versus latitudes. A linear regression between the tidal amplitude and geomagnetic dipole moment is
calculated. Figures 9a and 9b illustrate the estimated coefficient for the linear regression in the altitude-latitude plane, with
regions where the absolute value of the correlation coefficient is less than 0.6 being masked. The results show that as the
geomagnetic dipole moment increases per $10^{22}AM^2$ the thermospheric DW1-T in March would enhance by 1–3 K, with two
maximums around ±30°–40°. The response of SW2-T is much smaller and insignificant. At the equator, the increase in
geomagnetic dipole moment by $10^{22}AM^2$ would lessen the SW2-T amplitude merely ~0.3 K. A slight enhancement of SW2-T
due to the increase in geomagnetic dipole moment could be found in the upper thermosphere at mid-latitudes, while the growth
rate is only ~0.4 K/$10^{22}AM^2$.
**4 Discussion**
In this paper, two control runs, CR1 and CR2, were conducted to examine the response of thermospheric dynamics to long-
term changes in CO2 and geomagnetic field during the last 12,000 years of the Holocene. The CO2 effect was revealed as an





enhancement of the general circulation with increasing CO2 levels (Figure 1–2), which agreed with the result of Liu et al.
(2020). Rind et al. (1990) also found that an increase in CO2 similarly enhanced the mesospheric circulation. Both of them
suggested that the increased eddy forcing and gravity waves (GWs) should play an important role. However, the GCITEM-
IGGCAS model does not involve a parameterization scheme for GWs because the GWs mainly affect the mean flow in the
mesosphere rather than in the thermosphere. Therefore, the changes in the circulation caused by CO2 variations in our results
cannot be attributed to GWs. The interpretation by Kogure et al. (2022) should be responsible for the fact that the changes in
ion drag, molecular viscosity, and meridional pressure gradient forces are in the combined modulation. An interesting founding
is that the CO2 increase does not linearly accelerate the circulation and tends to be "saturated" as shown in Figure 3. The
plausible explanation is the molecular viscosity is non-linearly related to the temperature. As for the tidal response to the CO2
effect, the DW1 amplitude is positively correlated with CO2 variation (Figure 4). A reasonable deduction is that the decreased
viscosity due to the enhanced CO2 cooling should be less likely to dissipate tidal propagation from below. The latitudinal
structure of the DW1 response to CO2 differs from that of Liu et al. (2020), partly because their results mixed the influences
of changes in tidal sources from below, whereas our results reflected the internal thermospheric responses.

Figure 5 illustrated an asymmetric response in circulation to the geomagnetic variation. The change in neutral winds was larger
in the hemisphere with a more distant geomagnetic pole shift. Given the variation in the dipole component of the geomagnetic
field is hemispherical symmetric, it could logically infer that the hemisphere difference in circulation is contributed by the
variation of the non-dipole component. The neutral temperature change due to geomagnetic variation has a similar pattern to
the ion temperature in Cnossen et al. (2014), which is also manifested to decrease around the daytime equatorial ionization
anomaly (EIA) peaks. A possible causal linkage could be proposed that the geomagnetic variation affected the equatorial
plasma drift velocity, and then redistributed the electron density around the EIA region. As the electron density becomes
large/small the electron temperature changed conversely. The ion temperature change then should be more or less related to
the electron temperature change. Generally, the smaller strength of the geomagnetic fields would induce stronger equatorial
$\mathbf{E} \times \mathbf{B}$ drift and thus increase the electron density at the EIA peaks, and Yue et al. (2022) confirmed such a relationship. During
the nighttime, the equatorial drift tended to be downward and the EIA structure disappeared in general. So, the above-discussed
causality is not valid and the nighttime neutral temperature response should be different. The increased Joule heating related
to the weakening of the geomagnetic field might be responsible. Hence, the geomagnetic variation would redistribute the
temperature in the daytime and nighttime differently (Figure 6), then caused the day-night difference in Figures 6 and 7. The
seasonal dependence of the day-night difference in temperature response to the geomagnetic variation is still puzzled and needs
further explanation in the future. The temperature redistribution due to geomagnetic variation then causes the tidal responses
in Figures 8 and 9. At mid-latitudes, both DW1 and SW2 manifest to be positively correlated to the diploe moment, partly



because the strengthen geomagnetic field leading to the lower thermosphere (Cai et al., 2023) modulated the tidal propagation
from below. At the low-latitudes, the effect from $\mathbf{E}\times\mathbf{B}$ drift at daytime becomes important as aforementioned, therefore
different from that in mid-latitudes.

As a tentative investigation of the long-term change of thermospheric dynamics during ~12,000 yrs, this paper still has some
limitations and flaws, and one of them is the fixed lower boundary. In the present work, the migrating tides at the lower
boundary (90 km) are set to be unvaried regardless of simulating different periods in the Holocene. To our knowledge, the
long-term trend around mesopause is still debated, and the understanding changed from no trend to a mild negative trend in
general (Beig, 2003; Huang et al., 2014; Laštovička, 2017). This is partly because the temperature trends at these heights are
sensitive to the changes in stratospheric ozone concentration (Lübken et al., 2013). A whole atmosphere simulation performed
by Solomon et al (2018) also indicated there are very weak trends in the mesopause region. Hence, the perpetual lower
boundary should be a conservative and compromised treatment, additionally considering little evidences have been provided
on how the atmospheric tides change during such a long-term historical time. Besides, the fixed lower boundary inferred that
the tidal source from the lower atmosphere is constrained to be unvaried, so our results mainly describe the effect of
propagation conditions and local excitation on the long-term dynamics change in the thermosphere. In the next step, simulation
based on a whole atmosphere climate model, like the WACCM-X (Liu et al., 2018) and GAIA (Jin et al., 2011), should give
a much more realistic scenario of the long-term change in the thermospheric dynamics, nevertheless, the computation cost will
increase substantially.

In addition, the empirical model describing the high-latitude input, Weimer-96, is based on modern satellite measurements.
Although the geomagnetic intensity variation did not take into consideration, the effect of the geomagnetic tilted angle is
included in the model. The drift of magnetic poles and aurora region is thus considered given the Weimer-96 is based on a
magnetic coordinate. The intensity of the geomagnetic field is examined to influence the magnetosphere configuration and
thus expected to affect the energy input to the high-latitude thermosphere (Zhong et al., 2014; Cnossen et al., 2012). Vogt et
al. (2009) summarized the potential impact of the geomagnetic field variation on the geospace by modulating the shielding of
the energetic charged particles. During the simulated period, the dipole moment ($M$) is in the $6\times10^{22}$–$1\times10^{23}$ Am$^2$ range. As
the sine of polar cap size ($\theta$) is generally proportional to $M^{-1/6}$, a rough estimation deduces that $\theta$ would change by ~3°, within
latitudinal resolution (5°) in the model. Theoretical scaling about cross-polar cap potential ($\Phi$), $\Phi\propto M^{1/3}$, inferring that the $\Phi$
should varied from 18 to 21 kV during the Holocene if we set the $\Phi$ as 20 kV at the present era. Comparing a typical
geomagnetically disturbed condition that $\Phi$ is ~80 kV for Kp = 4, the relative change in $\Phi$ above is quite small. Cnossen et al.
(2014) also declared that the magnetosphere-ionosphere coupling only significantly during the disturbed conditions. Given our

https://doi.org/10.5194/egusphere-2023-234




simulation is perpetually geomagnetically quiescent, the impact of geomagnetic variation on the high-latitude energy input
should be limited.

In this work, the $CO_2$ and geomagnetic fields were regarded as two independent external driving to the simulation regardless
of their interaction, although whether the interaction exists is still controversial. Zhou et al. (2021) proposed that the decrease
in geomagnetic intensity would redistribute the $CO_2$ in the upper atmosphere using the whole atmosphere simulation. Their
investigation suggested that the increased ionospheric conductivities due to the weakened geomagnetic intensity would induce
much more Joule heating to warm the high-latitude lower thermosphere, which then should enhance the upwelling flow and
bring rich $CO_2$ from below. This result is based on the physical fact that the $CO_2$ distribution becomes deviated from the well-
mixed equilibrium above the mesopause (~80–90 km) and the time scale of eddy diffusion becomes much larger in the upper
atmosphere (Beagley et al., 2010; Rezac et al., 2015), so that the dynamical processes could modulate the $CO_2$ distribution.
However, up to date, little observational evidence has been proposed to support the possible link between $CO_2$ and
geomagnetic fields. A simulation project conducted by the whole atmosphere model in the next step could provide more
information.

Responses of the non-migrating tides to the variation of $CO_2$ and geomagnetic fields were not considered in this paper. The
eastward propagating diurnal tides with a zonal wave number of 3 (DE3) should be not much sensitive to the $CO_2$ change,
according to the discussion by Liu et al. (2020). This result was expected as the longitudinal variation of $CO_2$ concentration
is generally not obvious. On the other hand, geomagnetic fields crucially influence the non-migrating tidal propagation in the
upper atmosphere, through the electro-dynamo or parallel-line transport. For example, Jiang et al. (2018) revealed that DE3
tide can induce the longitudinal wavenumber-3 (WN3) structure rather than the should-be WN4 structure through the electro-
dynamical coupling with the geomagnetic field. Zhang et al. (2020) proposed that the significant role of parallel-line transport
alters the interhemispheric symmetry as the enhanced planetary waves upward propagated during the 2009 sudden stratosphere
warming (SSW) event. As the realistic geomagnetic field is much more complicated than the dipole or tilted dipole, a given
non-migrating tides propagating into the thermosphere would broaden the spectra of wavenumber. Yue et al. (2013) found that
there were complicated longitudinal structures rather than simply the WN3 as the quasi-2-day wave with westward zonal
wavenumber 3 propagating into the upper atmosphere. In this future work, the non-migrating tidal response to the long-term
variation will be worth studying.

10



**5. Conclusions**
This paper diagnosed the long-term changes in the thermospheric dynamics caused by the secular variation of $CO_2$ emissions
and geomagnetic field during the Holocene, using the global coupled thermosphere-ionosphere model, GCITEM-IGGCAS.
Two sets of long-term time-slice simulation covering ~12,000 yrs were performed by independently controlling the $CO_2$ level
and the configuration of geomagnetic fields, both under the perpetual condition of solar minimum and geomagnetic quiescence.
The corresponding changes in the circulation and major solar tides in the thermosphere were then analyzed, and the main
results were summarized as follows:
1. The $CO_2$ increase/decrease generally strengthened/weakened the general circulation in the thermosphere simultaneously,
and notably a dramatic strengthen in the circulation as the $CO_2$ steeply increases since the industrial revolution. The circulation
increase due to the $CO_2$ variation was examined to be non-linearly growth, which is expected to be caused by the nonlinear
relationship between temperature and molecular viscosity.
2. The amplitude of the diurnal migrating tide in the thermosphere will strengthen as the $CO_2$ increases throughout the
Holocene because the increased $CO_2$ cooling provides a plausible condition for tidal propagation.
3. Secular variation of geomagnetic field have a regional impact on the thermospheric circulation, particularly pronounced at
high latitudes and around the dusk sector. The prominent hemispheric differences in the thermospheric circulation response
infer a crucial role of the geomagnetic non-dipole component.
4. Geomagnetic variations also redistribute neutral temperature at mid- and low-latitudes and lead to different responses in the
daytime and nighttime, which then influence the thermospheric dynamics.
5. The geomagnetic dipole moment is highly correlated DW1 tidal amplitude at mid- and low-latitudes during March, and an
enhancement of $1 \times 10^{22}$ Am$^2$ will cause an increase in ~1–3 K of DW1-T in the thermosphere.
**Data availability**
The spherical harmonic coefficients of CALS10k.2 model was obtained from the website: https://earthref.org/ERDA/2207.
The IGRF model was downloaded from the website: https://www.ngdc.noaa.gov/IAGA/vmod/igrf.html. The Antarctica
Vostok and EPICA Dome C ice cores $CO_2$ level was derived from the website: https://data.noaa.gov/dataset/dataset/noaa-
wds-paleoclimatology-aicc2012-800kyr-antarctic-ice-core-chronology.The Antarctica Law Dome ice core $CO_2$ data was
downloaded from the website: https://www.ncei.noaa.gov/access/metadata/landing-page/bin/iso?id=noaa-icecore-9959. The
Mauna Loa observed $CO_2$ was from the website: https://gml.noaa.gov/ccgg/trends/data.html. The simulated data by GCITEM-
IGGCAS model under different control runs are available at: http://doi.org/10.17605/OSF.IO/ZQ8HY.



**Acknowledgments**
The authors acknowledge the support of the B-type Strategic Priority Program of the Chinese Academy of Sciences (Grant
XDB41000000), the Project of Stable Support for Youth Team in Basic Research Field, CAS (YSBR-018), the National
Natural Science Foundation of China (41621004, 42241106, 42204165), the CAS Youth Interdisciplinary Team (JCTD-2021-
05), and the Key Research Program of the Institute of Geology and Geophysics, CAS (Grant IGGCAS-201904).

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



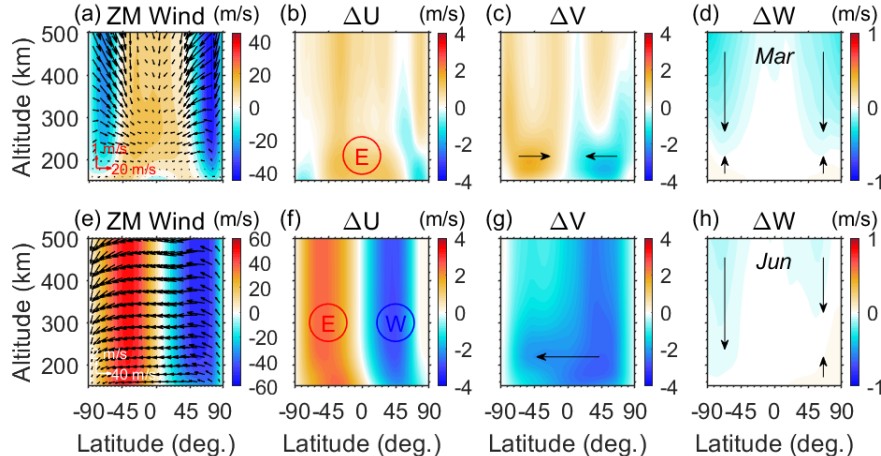

**Figure 1.** (a) Thermospheric circulation is illustrated by contours (zonal) and arrays (meridional and vertical) in March 2015. (b)–(d) Changes in zonal, meridional and vertical wind velocity due to the increase of CO2 from 1945 to 2015. Plots (e)–(f) are the same as plots (a)–(d) but for June.





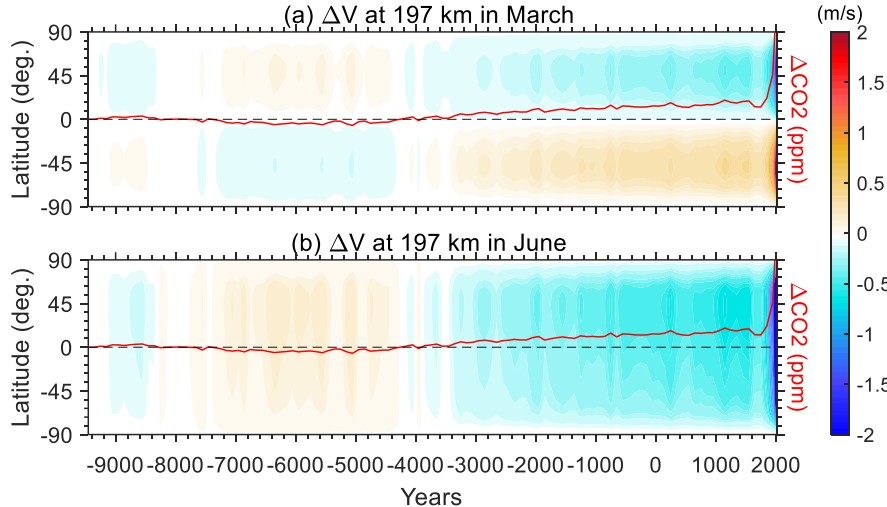

**Figure 2.** Time evolution of the changes in the zonal-mean meridional wind at 197 km during (a) March and (b) June. The corresponding CO2 variation is plotted in the red solid line.





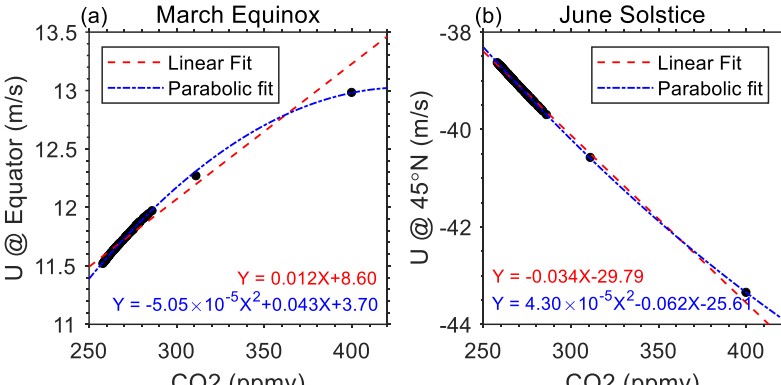


**Figure 3.** Response of thermospheric zonal-mean zonal winds (150–600 km average) to the CO2 increase (a) at the equator in the March equinox. (b) at 45°N in the June solstice. Linear and parabolic fitting are indicated in red-dashed and blue-dash-dotted lines, respectively.




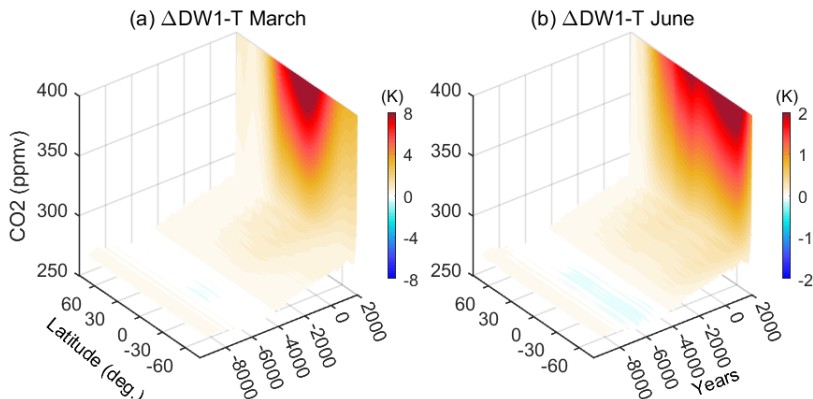


**Figure 4.** Change in the amplitude of diurnal migrating tide (DW1) at 240 km due to the CO2 variation in (a) March and (b)
June with respect to the beginning of the simulation.




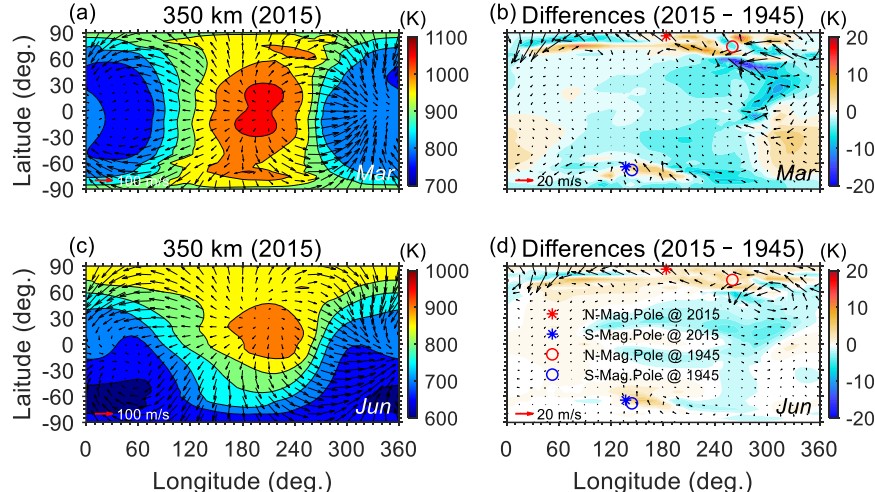

**Figure 5.** Geographic distribution of neutral temperature (color contours,) and horizontal winds (black arrows) at 350 km in (a) March and (c) June at UT00. (b) Differences in neutral temperature and horizontal winds due to changes in geomagnetic field between 1945 and 2015. The scales of wind velocity are indicated in the lower-left corner of each plot. The changes of north and south magnetic poles between 1945 and 2015 are illustrated in plots (b) and (d).





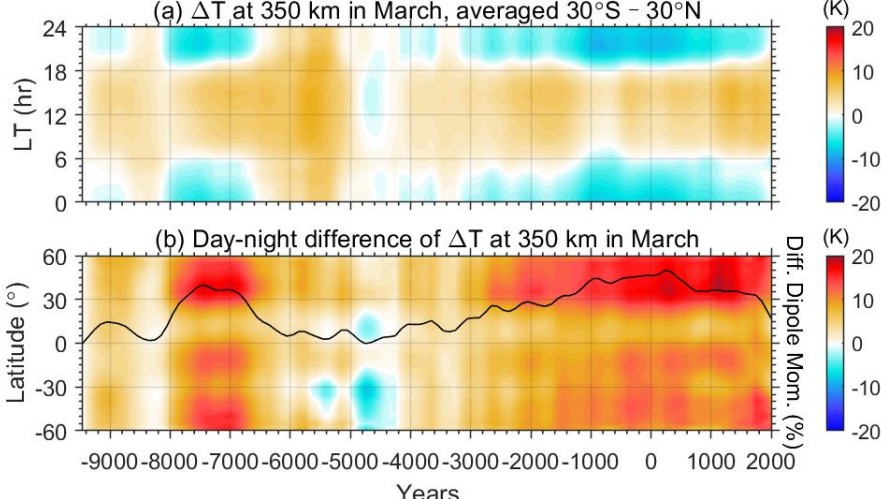

**Figure 6.** (a) Local-time (LT) variation of the zonal-mean temperature changes at low latitudes (30°S–30°N) caused by the secular variation of geomagnetic fields at 350 km in March during the Holocene. (b) Latitudinal variation of day-night differences in the zonal-mean temperature during March plotted versus year and with respect to the beginning of the simulation. The daytime and nighttime are corresponding to LT10–14 and LT22–02, respectively. Relative change of the geomagnetic dipole moment is plotted in the black-solid line in plot (b).



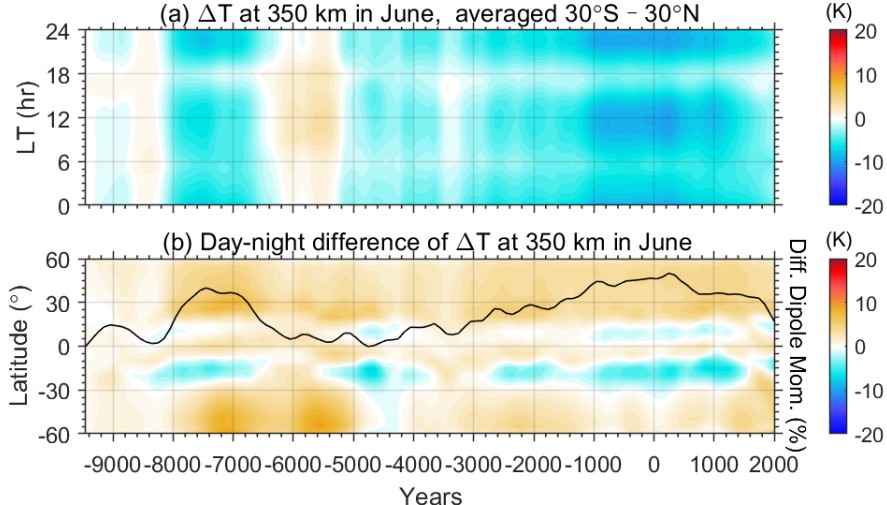

519

520 **Figure 7.** Same as Figure 6, but for the case of June.

521





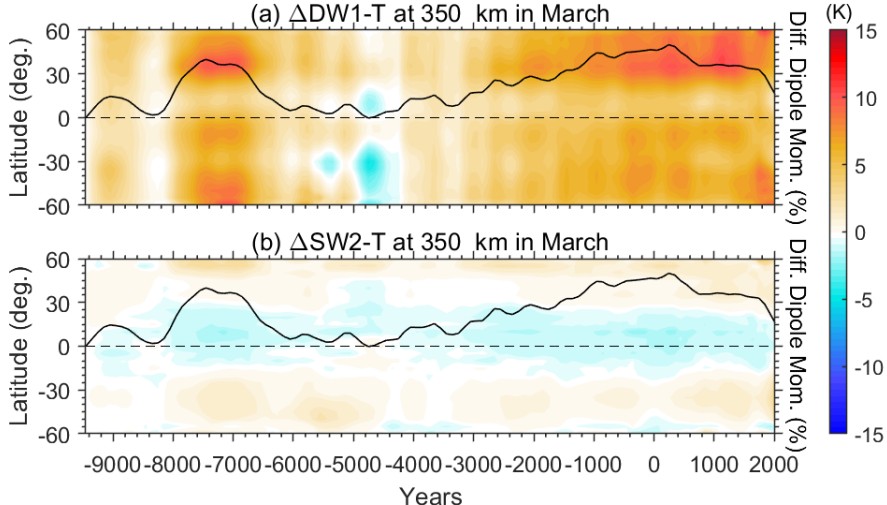

**Figure 8.** Time evolution of the differences in the amplitude of (a) DW1 and (b) SW2 with respect to the beginning of the simulation.



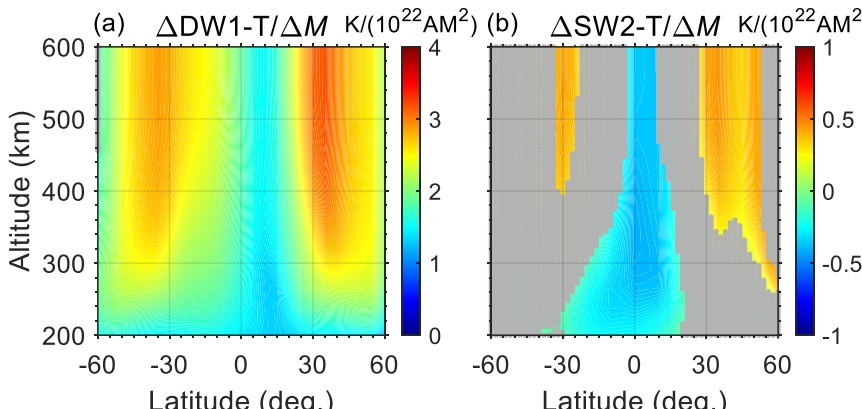

**Figure 9.** Regression of (a) DW1-T and (b) SW2-T amplitudes on the geomagnetic dipole moment. The grey shaded area indicates where the absolute values of correlation coefficients are less than 0.6.