# Peer review of "Simulated Long-term Evolution of the Thermosphere during the"

_EGUsphere, 2023_

## Author Response (AR1)

We thank both referees for his/her valuable comments that have helped to improve the quality and clarity of the manuscript. We have tried our best to address all the comments point-to-point as detailed in the following.

**Referee #1**

*\* In Figure 2: Why is it opposite the changes in V between the northern and the southern hemisphere? Maybe I am missing something here.*

**Response:** Thanks for the comment. The reason for the opposite change in circulation in the northern and southern hemispheres during the equinox is that the mean flow becomes more poleward, and we define the positive V as northward. Thus, the change in V is manifested as positive in the northern hemisphere and negative in the southern hemisphere.

*\* Figure 6: Why average between +30 and -30? That is the region enclosing the magentic equator, so there can be different variations in each hemisphere, so they will eventually cancel.*

**Response:** Thanks for the comment. We did the average exactly to eliminate the interhemispheric differences, so could focus on the effect of geomagnetic field variations on the low-latitude day-night differences in the mean state.

*\* What about the variation of the magnetic equator (or geomagnetic equator) along the holocene. Because I understand that you are considering an almost dipolar field, but not axial. So the changes in its tilt (and displacement form the center in addition) would contribute to the equator shifts.*

**Response:** Thanks for the comment about the variation of magnetic equator. Changes of magnetic equator are an important factor in the ionosphere-thermosphere system indeed. For example, the EIA or ETA structure would shift according to the tilt change at a given longitude. Figure 1 of Cai et al. (2023), which is the companion of this paper, presented the effect of magnetic field on the thermosphere. However, when discussing the changes in thermospheric dynamics at mid- and low-latitudes, we found the effect of dipole change is more dominant.

*\* Line 34: I think that in "since the Holocene", it may should be "since the beginning of the Holocene", or something like this.*

**Response:** Thanks for the suggestion, and we have revised this sentence like this.

\* Line 56: in "The geomagnetic tiled angle ..." I think tiled has a typo error. I think it may be "tilt" or linked to this ?

**Response:** Thanks for pointing out the typo, and we have corrected them in the manuscript.

\* Line 111: in "the thermospheric dynamical response.." there are two dots at the end. Delete one.

**Response:** Thanks for the careful examination, and we have deleted the surplus comma.

\* Line 118: in "equatorial meridional winds winds" delete one "winds"

**Response:** Thanks for the suggestion. We are sorry for this typo and we have deleted it.

\* Line 136: About the fluctuations. Why one pair of values have minus-plus (+-) and the other just minus (-)?

**Response:** Thanks for the comment. The minus-plus indicated the V change is different in the northern and southern hemisphere in March, while it is the same during June.

\* Line 166: "The circulation change in the northern hemisphere is much larger in the southern hemisphere, regardless during March or June." I do not understand this sentence. Maybe you want to say that the circulation change was larger in the northen than in the southern hemisphere? And what about "regardsless during March or June? You mean that in the two months it is observed this relative difference between north and south?

**Response:** Thanks for the comment. We do intend to express that the circulation change in the northern hemisphere is greater than the southern hemisphere in different seasons. We are sorry for the misleading and revised the sentence as follows:

"The circulation change shows a larger change in the northern hemisphere than in the southern hemisphere, in both simulations for March and June."

\* Caption of Figure 1: "Thermospheric circulation is illustrated by contours (zonal) and arrays (meridional and vertical)". What do you mean by "arrays"? Maybe you mean "arrows"?

**Response:** Thanks for the careful check, and we have corrected the mistake.

\* Figure 9. Something is wrong with the values in the colorbar of the left figurre. Shouldn't they be between 1 and -1?

**Response:** Thanks for the comment. The values in Figure 9 are not the correlation coefficients, but the coefficient estimates for the linear regression. Hence, we have revised the caption to avoid misunderstanding as follows:

"Coefficient estimates for the linear regression of (a) DW1-T and (b) SW2-T amplitudes on the geomagnetic dipole moment. The grey shaded area indicates where the absolute values of correlation coefficients are less than 0.6."

**Referee #2**

*Lines 51-63: I recommend add sentence "The secular changes of geomagnetic field produce regionally both positive and negative changes, therefore in the global average their effect is negligible (Qian et al., 2021).*

*Qian, L., McInerney, J. M., Solomon, S. S., Liu, H., and Burns, A. G.: Climate changes in the upper atmosphere: Contributions by the changing greenhouse gas concentrations and Earth's magnetic field from the 1960s to 2010s, J. Geophys. Res. Space Phys., 126(3), e2020JA029067, https://doi.org/10.1029/2020JA029067, 2021.*

**Response:** Thanks for the suggestion. We have added this sentence and reference in the manuscript.

*In Section 2 you describe scenario CR1 but not CR2 (only reference). Please add for comfort of readers a very brief description of scenario CR2.*

**Response:** Thanks for the comment. We have added the following sentences to describe the CR2 scenario in the manuscript:

"…The secular variation of geomagnetic field implemented in the CR2, including the dipole moment and the position of magnetic and geomagnetic pole, was illustrated in Figure 1 of Yue et al. (2022), and the readers could refer to Constable et al. (2016) for more detailed information. A general scenario includes: a) the dipole moment fluctuated within 6.1–10.1 (1022AM2), and has continuously decreased since 1700 by ~13%. b) The geomagnetic/magnetic pole located at latitudes larger than 78°/70°, and drifted from the western hemisphere to the eastern hemisphere over the past century…"

*Lines 235-236: weaker B induces stronger E x B - ? – purely mathematically weaker B should induces stronger E x B.*

**Response:** Thanks for the comment. Assuming the E-field unchanged, the inverse relationship is generally valid as the drift velocity is $ExB/B^2$. Hence, we revised the manuscript to avoid the misleading as following:

"…Generally, the smaller strength of the geomagnetic fields would induce stronger equatorial vertical drift ($E×B/B^2$) and thus increase the electron density at the EIA peaks…"

*Section Discussion, flaws: Another potential flaw: According to GAIA simulations by Liu et al. (2021) the efficiency of CO2 forcing somewhat differs under low and high geomagnetic activity conditions.*

*Liu, H., Tao, C., Jin, H., and Abe, T.: Geomagnetic activity effect on CO2-driven trend in the thermosphere and ionosphere: Ideal model experiments with GAIA. J. Geophys. Res. Space Phys., 126(1), e2020JA028607, https://doi.org/10.1029/2020JA028607, 2021.*

**Response:** Thanks for the comment. We have added some discussion about this point as following:

"In addition, this paper only considered the geomagnetically quiet condition, while the efficiency of CO2 forcing somewhat differs under low and high geomagnetic activity conditions according to GAIA simulations by Liu et al. (2021)."

*Wording and misprints:*
*Line 42: some is missing in the sentence – it does not make sense*
*Line 98: change to "interval from ~80,000 to ~1650 yrs. before present"*

*Line 134: "varied to be much equatorward/poleward" should be "tends to be more/less equatorward"*

*Line 153: "than that" should be "of that"*

*Line 166: "larger in" should be "larger than in"*

*Line 185: This sentence is not understandable.*

*Line 207: "merely" should be "merely by"*

*Line 221: "is the molecular" should be "is that the molecular"*

*Line 243: "diploe" should be "dipole"*

*Line 244: something is missing, the sentence does not make sense*

*Line 272: "significantly" should be "significant"*

*Line 311: sentence is not understandable*

*Line 312: "was examined to be non-linearly growth" should be "was found to grow non-linearly"*

*Line 321: "correlated DW1" should be "correlated with the DW1"*

*Line 488: "contours" should be rather "colors"*

**Response:** Thanks for the careful check. We have corrected the mistakes pointed by the referee.